# Prevalence of Alpha(α)-Thalassemia in Southeast Asia (2010–2020): A Meta-Analysis Involving 83,674 Subjects

**DOI:** 10.3390/ijerph17207354

**Published:** 2020-10-09

**Authors:** Lucky Poh Wah Goh, Eric Tzyy Jiann Chong, Ping-Chin Lee

**Affiliations:** Biotechnology Programme, Faculty of Science and Natural Resources, Universiti Malaysia Sabah, Kota Kinabalu 88400, Sabah, Malaysia; luckygoh@hotmail.com (L.P.W.G.); eric_ctj@live.com (E.T.J.C.)

**Keywords:** prevalence, α-thalassemia, Southeast Asia, meta-analysis, haematological disorder

## Abstract

Alpha(α)-thalassemia is a blood disorder caused by many types of inheritable α-globin gene mutations which causes no-to-severe clinical symptoms, such as Hb Bart’s hydrops fetalis that leads to early foetal death. Therefore, the aim of this meta-analysis was to provide an update from year 2010 to 2020 on the prevalence of α-thalassemia in Southeast Asia. A systematic literature search was performed using PubMed and SCOPUS databases for related studies published from 2010 to 2020, based on specified inclusion and exclusion criteria. Heterogeneity of included studies was examined with the I2 index and Q-test. Funnel plots and Egger’s tests were performed in order to determine publication bias in this meta-analysis. Twenty-nine studies with 83,674 subjects were included and pooled prevalence rates in this meta-analysis were calculated using random effect models based on high observed heterogeneity (I2 > 99.5, *p*-value < 0.1). Overall, the prevalence of α-thalassemia is 22.6%. The highest α-thalassemia prevalence was observed in Vietnam (51.5%) followed by Cambodia (39.5%), Laos (26.8%), Thailand (20.1%), and Malaysia (17.3%). No publication bias was detected. Conclusions: This meta-analysis suggested that a high prevalence of α-thalassemia occurred in selected Southeast Asia countries. This meta-analysis data are useful for designing thalassemia screening programs and improve the disease management.

## 1. Introduction

Thalassemia is the most common hereditary red blood cell disorder which causes anemias due to defective genes that code for globin proteins synthesis [1]. The inheritance of the thalassemia genotype could result in the individual being either a carrier or a patient. There are two major types of thalassemia: (1) alpha (α) and (2) beta (β), in which the former is the most common form of thalassemia worldwide especially in Southeast Asia populations [2,3]. Both α- and β-thalassemia arise from genetic defects in α and/or β-globin genes, which regulate the number of globin chains in red blood cells. Genetic defects in either α and/or β-globin genes can cause imbalance in numbers of α and β chains in red blood cells. This leads to the manifestation of clinical conditions known as α- or β-thalassemia. The most common genetic defect in α-thalassemia is a deletion in the α-globin gene involving one or both globin genes such as -α^3.7^, -α^4.2^, --^SEA^, --^THAI^, -α^CD59^, -α^20.5^, -α^IVS I-1^ and others (Figure 1) [4,5,6]. A highly severe form of deletional α-thalassemia, known as Haemoglobin (Hb) Bart’s hydrops fetalis, is a homozygous α^0^-thalassemia deletion with a complete loss of functional α-globin that leads to foetal death or death shortly after birth. Currently, there is no effective treatment for this disease [7,8].

There are also cases of non-deletional mutations in the α-globin gene such as Hb Constant Spring (CS), which is caused by a nucleotide substitution in the termination codon TAA→CAA and also Hb Pakse (-α^4PS^), which is caused by the termination codon (UAA→CAA). This results in the elongation of the α-globin chain protein and causes severe anemia with serious complications that include liver impairment, cardiac disease and endocrine disorder [9,10]. Some cases of Hb CS require red blood cell transfusions when Hb drop to dangerously low levels [11]. Most importantly, an individual can carry a single or multiple type form of the mutation (deletional/non-deletional), which give rise to different clinical manifestations and complicates diagnosis as well as treatments for α-thalassemia [12]. Both deletional and non-deletional α-thalassemia prevalence rates are highly important in determining the overall severity of clinical symptoms in a region or a specific population.

Despite the detrimental effects of thalassemia, evidence shows that thalassemia confers a protective effect against hyperparasitemia due to malaria infection [13]. *Plasmodium vivax* parasitemia were two to three times lower in thalassemia patients as compared to malaria cases in people without thalassemia. However, the protective effect of thalassemia against parasitemia was not observed in a study conducted in Papua New Guinea in children aged 3–21 months [14]. Therefore, there is a natural selection force which leads to the prevalence of thalassemia cases in malaria-endemic areas [15].

Most of the α-thalassemia cases reveal some abnormalities in their red cell index. According to the British Committee for Standards in Haematology, a value of <27 pg in the average amount of Hb found in red blood cells (also referred as Mean Corpuscular Hemoglobin) is the primary screening threshold to quantify Hb subtypes [16]. However, subjects with a single gene deletion or carriers of the mutation in the non-severe form of α-thalassemia may present a normal Hb level. Different heterozygosity or homozygosity of gene deletions or mutations in the α-thalassemia gene gives different phenotypes, which complicates treatment [12]. Therefore, diagnosing of non-severe forms of α-thalassemia is a highly challenging task.

Several countries in the Southeast Asia region have reported the prevalence of α-thalassemia in different ethnic groups independently, revealing that the prevalence of α-thalassemia differs from country to country with different ethnicities [4,5,17,18]. However, no studies have systematically meta-analyzed the prevalence and epidemiology of α-thalassemia—where the results of these similar studies are quantitatively combined—for this region. Therefore, the aim of this meta-analysis was to provide an update (from 2010 until 2020) using data concerning α-thalassemia prevalence in Southeast Asia, focusing on Cambodia, Laos, Malaysia, Thailand, and Vietnam. The study outcome on the perspective of α-thalassemia prevalence in Southeast Asia could aid in designing healthcare policies for α-thalassemia screening in large populations and provide better genetic counselling programs.

## 2. Materials and Methods 

### 2.1. Study Guidelines and Literature Search

PRISMA guidelines were followed for conducting and reporting the results of this meta-analysis (Appendix A) [19]. PubMed (Appendix A) and SCOPUS (Appendix A) databases were searched up to March 2020 with the lower limit set to January 2010 using related terms, including alpha, thalassemia, southeast, and Asian.

### 2.2. Selection of Studies and Data Extraction

All search results were screened by two investigators, and all potential studies were independently reviewed to be included in this meta-analysis. The main inclusion criteria were: (1) studies published in English in which the prevalence of α-thalassemia (including all deletional and non-deletional mutations) in Southeast Asian countries were reported; and (2) the study was a peer-reviewed publication. Studies that did not report cross-sectional, observational, cohort, or prevalence of α-thalassemia were excluded. We identified additional eligible studies based on references that were cited in the relevant articles. When publications overlapped, only the study with the largest or the most recent data was included in this meta-analysis. Data, including first author’s name, publication year, country, sample size, and prevalence of α-thalassemia of the included studies, were extracted and documented by the reviewing investigators. A total number of 278 articles were included in this meta-analysis (Appendix A). The study selection and review process are illustrated in Figure 2.

### 2.3. Statistical Analyses

The prevalence of α-thalassemia was calculated for each study with the number of reported α-thalassemia cases as the numerator and the total sample size as the denominator. Homogeneity across studies was investigated with the I^2^ index (represented as percentage) and Q-test (represented as a *p*-value) that indicated heterogeneity between studies. It was reported that an I^2^ value > 75% and Q-test with a *p*-value < 0.1 was regarded as high heterogeneity [20,21]. A random effects model was used to combine individual effect sizes to create pooled α-thalassemia prevalence if a significantly high heterogeneity was observed. If other results were obtained, a fixed effects model was utilized. A forest plot was generated to illustrate the prevalence of each study with a 95% confidence interval (95% CI) that contributed to the analysis along with the combined prevalence rate. A subsequent meta-analysis was also performed based on each respective country. Funnel plots and Egger’s tests of asymmetry were performed to identify any bias within the results [22,23]. All analyses were performed with Comprehensive Meta-Analysis version 2 software (Biostat, Inc., New Jersey, USA) [24].

## 3. Results

### 3.1. Study Characteristics

Twenty-nine studies with a total number of 83,674 subjects were included in this meta-analysis after a detailed assessment of records obtained from the database and additional searching. These studies were published between January 2010 and October 2019. Among all included studies, two were carried out in Cambodia, three in Laos, five in Malaysia, 20 in Thailand, and two in Vietnam (Table 1). The main characteristics of the studies included in the meta-analysis were recorded and are shown in Table 1.

### 3.2. Meta-Analysis Outcomes

The meta-analysis was conducted using a random effects model found significant heterogeneity showed an I^2^ > 99.5% and *p*-value < 0.001 in overall and all subgroups except Cambodia (Table 2). The forest plot showed that the overall prevalence rate of α-thalassemia occurrence in this meta-analysis was 0.226 (95% CI = 0.168–0.296; I^2^ = 99.5%; *p*-value < 0.1) (Figure 3). In the subgroup analysis based on country, Vietnam had the highest prevalence rate (51.5%) of α-thalassemia followed by Cambodia (39.5%) Laos (26.8%), Thailand (20.1%), and Malaysia (17.3%) (Figure 4).

### 3.3. Publication Bias

Funnel plots and Egger’s tests were performed to estimate the publication bias of the included literature. The shape of the funnel plot revealed obvious evidence of symmetry (Figure 5). The value for Egger’s test was *t*-value = 1.24 with a *p*-value = 0.112.

## 4. Discussion

This study is the first to report the prevalence of α-thalassemia in the Southeast Asia region over the past 10 years (2010–2020). We did not obtain any α-thalassemia-related studies that fulfilled our inclusion and exclusion criteria from other Southeast Asian countries, including Brunei, Indonesia, Myanmar, Philippines, Singapore, and Timor-Leste. Using a random effects model, the overall prevalence rate of α-thalassemia in the included countries was 22.6%, which indicated a significant reduction of ~50% of the prevalence in the Southeast Asia region since 2008. The World Health Organization had reported the α-thalassemia prevalence as 44.6% in 2008 [49]. India and Brazil, were reported at about 12% and 19.2%, respectively [50,51]. The prevalence rates were high in countries such as UAE, Oman and Saudi Arabia at 50% [52].

The high prevalence of α-thalassemia in the past was due to the lack of knowledge regarding the seriousness of this disease among the populations in these countries, especially those living in rural areas with limited access to education and those who could not afford to obtain the proper education similar to that found in urban areas. α-thalassemia is an inherited disease and the mutations may pass from parent to child, affecting the haemoglobin production. Hence, the educational and screening campaigns regarding this disease conducted by the representative bodies (either government or non-government organization) have successfully reduced the prevalence of α-thalassemia in the Southeast Asia region.

Random effects models were used, which are based on the assumption that the true effect could vary between studies [51]. The existence of publication bias in this meta-analysis was determined using a funnel plot and Egger’s tests. The shape of the funnel plot in this meta-analysis showed an obvious symmetry, indicating the risk of publication bias is significantly low. This hypothesis was also supported by statistical evidence from Egger’s test (*t*-value = 1.24; *p*-value = 0.112) in which the publication bias did not significantly exist in this meta-analysis. Therefore, we concluded that there was no publication bias detected in this meta-analysis.

When stratified according to country, Vietnam has the leading α-thalassemia prevalence rate of 51.5%. The high prevalence rate is probably due to one of the observational studies conducted in Vietnam focusing on the country’s minority ethnic groups and thus, this likely skewed the actual prevalence rate [51]. The prevalence of α-thalassemia in Cambodia was the second highest (39.5%) when compared to other countries included in this meta-analysis. However, since there was only two studies that reported the prevalence of α-thalassemia Cambodia and Vietnam, more data are required to estimate the actual prevalence rate of this disease in both of these countries. The prevalence of α-thalassemia in Laos, Malaysia, and Thailand was quite similar, ranging from 17.3% to 26.8%. Alpha thalassemia is an inheritable disease where the presence of multiple deletional and non-deletional mutations can cause severe clinical complications. The presence of α-thalassemia major causes hydrops fetalis and prenatal deaths [4,5]. Therefore, the low prevalence of α-thalassemia in these countries and this region indicates that genetic screening for α-thalassemia mutations in the parents could be done in a population focused approach. Allele frequency and genetic diversity amongst the different populations provide information that can be used effectively in designing thalassemia prevention programs [53].

Thalassemia patients suffers from anemia caused by the imbalance of globin chains and impairment of haemoglobin solubility of erythrocytes. The reduced globin chains were shown to impair the cytoadherence of *Plasmodium* [54,55]. These abnormalities of erythrocytes have been shown to confer a protective effect against malaria infection [12,56]. Hence, there is a natural selection pressure which causes thalassemia becoming prevalent in countries with high incidence of malaria. Our meta-analysis was not exhaustive, however it shows that Vietnam had the highest prevalence rate (51.5%) and the lowest malaria cases (5794 cases) among the countries included in our study which supported the protective factor of thalassemia [57].

There are several limitations that should be addressed in this meta-analysis. Firstly, only data from certain Southeast Asian countries were available to be included in this meta-analysis; therefore, the calculated α-thalassemia-related prevalence rate in this study was specific to selected Southeast Asian regions. Besides, only studies from 2010 to 2020 were included in this meta-analysis, and it is possible for studies published before the year 2010 that meet the inclusion criteria but were not included in this meta-analysis, as this study focused on the prevalence rates from recent past 10 years only. We also did not include ethnic stratification because the majority of the studies included in this analysis did not report the ethnic group in the subject population. Alpha thalassemia genotype stratification was not be performed due to inconsistencies in the reporting of genotypes in the studies included in this analysis.

## 5. Conclusions

This is the first meta-analysis that investigated α-thalassemia prevalence in Southeast Asian countries, and the findings suggest high prevalence of α-thalassemia events in certain countries which warrants attention as α-thalassemia major could cause severe health complications and impose a substantial burden to the health authority and families. The data in this meta-analysis may be beneficial to the representative bodies in designing educational and screening campaigns regarding this disease in order to further reduce α-thalassemia rates in these countries.

## Figures and Tables

**Figure 1 ijerph-17-07354-f001:**
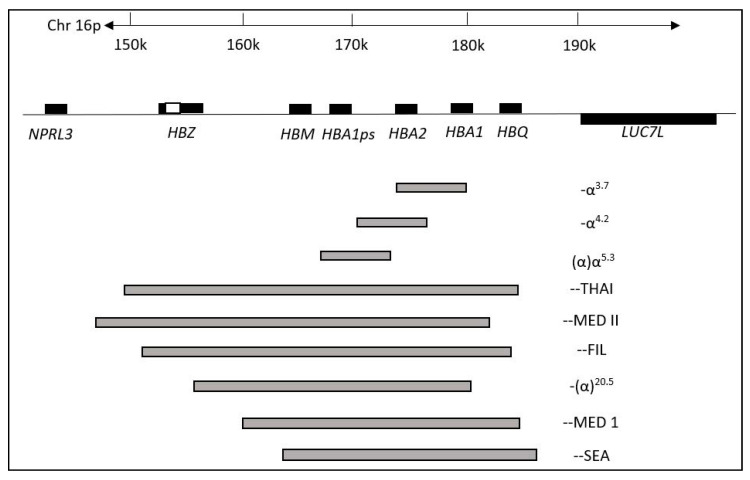
Some common changes in α-thalassemia in Southeast Asia. The genes are shown in boxes with a scale in kilobases (kb). The most common deletions of α-thalassemia mutations are indicated by grey bars indicating the length of deletion. (Adapted from Farashi & Harteveld, 2018 [6]).

**Figure 2 ijerph-17-07354-f002:**
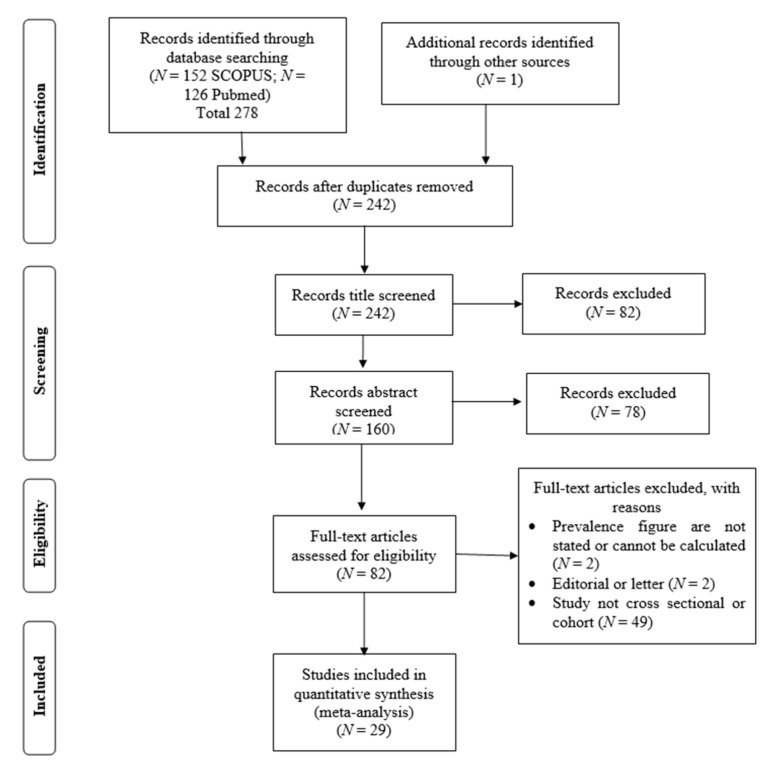
Flow diagram of the systemic literature search in this study.

**Figure 3 ijerph-17-07354-f003:**
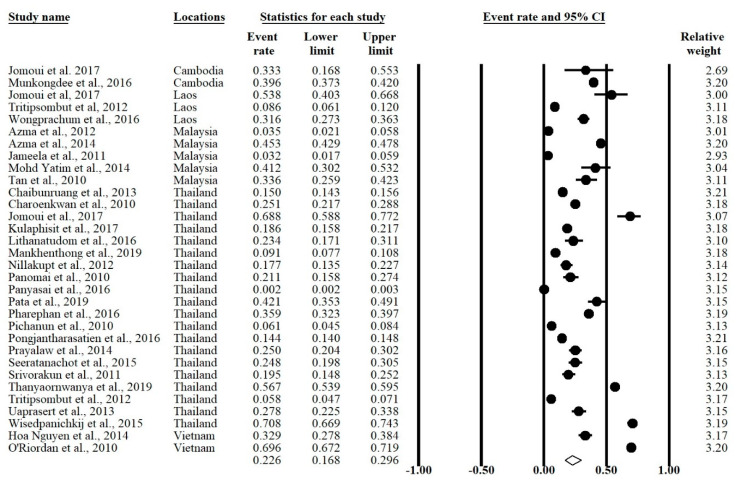
Forest plot of α-thalassemia overall prevalence using random effects model.

**Figure 4 ijerph-17-07354-f004:**
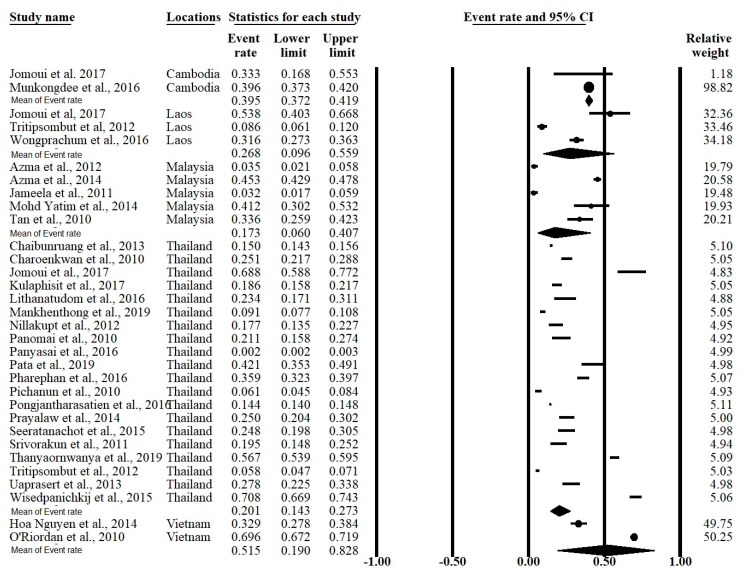
Forest plot of the α-thalassemia prevalence grouped according to country.

**Figure 5 ijerph-17-07354-f005:**
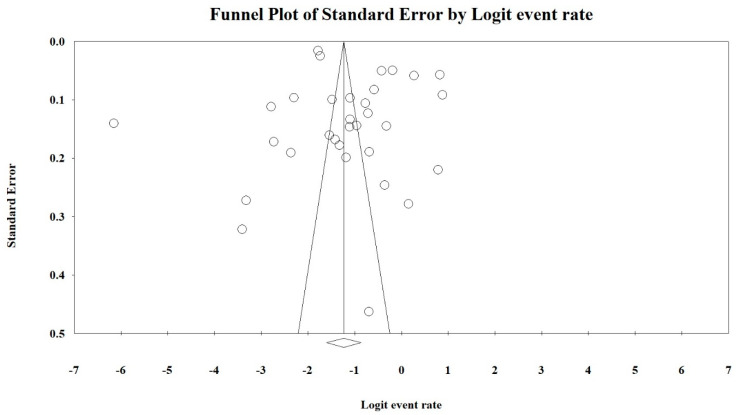
Funnel plot of the overall prevalence of α-thalassemia in this study.

**Table 1 ijerph-17-07354-t001:** Characteristics of studies included in the meta-analysis.

Author [Reference]	α-Thalassemia Genotyping Method	Genotypes Found in the Study	Country	Specific Ethnic ^1^	Events ^2^	Total ^3^
Munkongdee et al., 2016 [25]	Polymerase chain reaction (PCR)	-α^3.7^, -α^4.2^, --^SEA^, α^CS^, α^Ps^	Cambodia	N/A	646	1631
Jomoui et al., 2017 [26]	PCR	--^SEA^	Cambodia	N/A	7	21
Wongprachum et al., 2012 [27]	PCR	-α^3.7^, -α^4.2^, --^SEA^, --^THAI^, α^CS^, α^Ps^, α^Q-Thailand^	Laos	N/A	130	411
Jomoui et al., 2017 [26]	PCR	--^SEA^	Laos	N/A	28	52
Tritipsombut et al., 2012 [28]	PCR	-α^3.7^, -α^4.2^, --^SEA^, α^CS^, α^Ps^	Laos	N/A	30	349
Azma et al., 2012 [29]	PCR		Malaysia	N/A	14	400
Azma et al., 2014 [4]	PCR	-α^3.7^, -α^4.2^, --^SEA^, α^CS^, α^CD59^, α^IVS I-1^	Malaysia	Malay, Chinese, Indian, Other	736	1623
Jameela et al., 2011 [30]	PCR	-α^3.7^, -α^4.2^, --^SEA^, --^FIL^, α^125^	Malaysia	Malay, Chinese, Indian, Sikh, Iban	10	310
Mohd Yatim et al., 2014 [31]	PCR	-α^3.7^, --^SEA^, α^CS^, α^CD59^,	Malaysia	Malay	28	68
Tan et al., 2010 [3]	PCR	-α^3.7^, -α^4.2^, --^SEA^, --^THAI^, --^FIL^, α^CS^, α^125^,	Malaysia	Kadazandusun	42	125
Charoenkwan et al., 2010 [32]	PCR	-α^3.7^, -α^4.2^, --^SEA^, -α^Q-Thailand^, α^CS^	Thailand	N/A	142	566
Lithanatudom et al., 2016 [17]	PCR	-α^3.7^, -α^4.2^, --^SEA^, --^THAI^, α^CS^, α^Ps^	Thailand	Yong, Yuan, Lue, Khuen, Blang, Mon, Paluang, Lawa	33	141
Nillakupt et al., 2012 [33]	PCR	-α^3.7^, --^SEA^, α^CS^, α^Ps^	Thailand	N/A	47	266
Pongjantharasatien et al., 2016 [34]	PCR	--^SEA^, --^THAI^, --^FIL^, -α^thal-1^	Thailand	N/A	4555	31,632
Pichanun et al., 2010 [35]	PCR	-α^3.7^, α^CS^, α^Ps^	Thailand	N/A	36	587
Pharephan et al., 2016 [36]	PCR	-α^3.7^, -α^4.2^, --^SEA^, α^CS^	Thailand	N/A	229	638
Panyasai et al., 2016 [37]	PCR	-α^3.7^, -α^QT^, --^SEA^,	Thailand	N/A	51	23,914
Panomai et al., 2010 [38]	PCR	-α^3.7^, -α^4.2^, --^SEA^, --^THAI^, α^CS^, α^Ps^	Thailand	N/A	40	190
Prayalaw et al., 2014 [39]	PCR	-α^3.7^, -α^4.2^, --^SEA^, α^CS^, -α^Q-Thailand^	Thailand	N/A	75	300
Seeratanachot et al., 2015 [40]	Realtime-PCR	-α^3.7^, -α^4.2^, --^SEA^	Thailand	N/A	62	250
Wisedpanichkij et al., 2015 [41]	PCR	-α^3.7^, -α^4.2^, --^SEA^, α^CS^	Thailand	N/A	409	578
Uaprasert et al., 2013 [42]	PCR	-α^3.7^, -α^4.2^, α^CS^	Thailand	N/A	67	241
Srivorakun et al., 2011 [43]	PCR	-α^3.7^, --^SEA^, α^CS^	Thailand	N/A	44	226
Tritipsombut et al., 2012 [28]	PCR	-α^3.7^, -α^4.2^, --^SEA^, α^CS^, α^Ps^	Thailand	N/A	85	1460
Chaibunruang et al., 2013 [44]	PCR	--^SEA^, --^THAI^	Thailand	N/A	1874	12,525
Kulaphisit et al., 2017 [5]	PCR	-α^3.7^, -α^4.2^, --^SEA^, --^THAI^, α^CS^, α^Ps^	Thailand	Yong, Lue, Yuan, Shan, Khuen, Htin, Paluang, Blang, Lawa, Mon, Skaw Karen, Pwo Karen, Padong Karen	124	668
Thanyaornwanya et al., 2019 [45]	PCR	-α^3.7^, -α^4.2^, α^CS^, α^Ps^	Thailand	N/A	676	1192
Jomoui et al., 2017 [26]	PCR	--^SEA^	Thailand	N/A	66	96
Mankhenthong et al., 2019 [46]	PCR	-α^3.7^, -α^4.2^, --^SEA^, --^THAI^, α^CS^	Thailand	N/A	118	1290
Pata et al., 2019 [47]	PCR	-α^3.7^, -α^4.2^, --^SEA^, --^THAI^, α^CS^	Thailand	N/A	82	195
O’Riordan et al., 2010 [18]	PCR	-α^3.7^, -α^4.2^, --^SEA^, --^THAI^, --^FIL^, α^CS^	Vietnam	Kinh, Dao, Tay, Nung, S’Tieng, M’Nong, Rac Iay, E De	996	1431
Hoa Nguyen et al., 2014 [48]	PCR	-α^3.7^, -α^4.2^, --^SEA^, --^THAI^, --^SEA^, α^CS^, α^Ps^	Vietnam	Cό-Tu	98	298
Total					11,580	83,674

^1^ N/A: Not available due to ethnicities were not reported by the study. ^2^ Events: Number of subjects carrying alpha-thalassemia. ^3^ Total: Total number of subjects.

**Table 2 ijerph-17-07354-t002:** Prevalence rate and heterogeneity of α-thalassemia in overall and subgroups of the study.

Heterogeneity		Prevalence Rate (95% CI)	Sample Size (*N*)	No. of Studies (*N*)	Subgroups
I^2^ (%)	*p*-Value				
99.53	<0.001	0.226 (0.168–0.296)	83,674	32	Overall
0	0.560	0.395 (0.372–0.419)	1652	2	Cambodia
97.26	<0.001	0.268 (0.096–0.559)	812	3	Laos
98.20	<0.001	0.173 (0.060–0.407)	2526	5	Malaysia
99.47	<0.001	0.201 (0.143–0.273)	76,955	20	Thailand
99.22	<0.001	0.515 (0.190–0.828)	1729	2	Vietnam

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
