# Peer review of "Prevalence of Alpha(α)-Thalassemia in Southeast Asia (2010–2020): A Meta-Analysis Involving 83,674 Subjects"

_ijerph, 2020, doi:10.3390/ijerph17207354_

Round 1

Reviewer 1 Report

The manuscript entitled "Prevalence of alpha(α)-thalassemia in southeast Asia
3 (2010-2020): A meta-analysis involving 83,674 subjects" by Goh et al provides an extensive survey of literature on prevalence of α-thalassemia in Southeast Asian nations. 

The manuscript analyzed data from numerous studies from past ten years and included more than 83,000 subjects. However, the shortcoming of the manuscript is that authors can provide more background and significance about why the study was performed. Authors can expand the discussion and conclusions in detail to actually include the clinical relevance of their results.

Author Response

Reviewer 1

The manuscript entitled "Prevalence of alpha(α)-thalassemia in southeast Asia
3 (2010-2020): A meta-analysis involving 83,674 subjects" by Goh et al provides an extensive survey of literature on prevalence of α-thalassemia in Southeast Asian nations. 

The manuscript analyzed data from numerous studies from past ten years and included more than 83,000 subjects. However, the shortcoming of the manuscript is that authors can provide more background and significance about why the study was performed. Authors can expand the discussion and conclusions in detail to actually include the clinical relevance of their results.

Response: The author would like to thank the reviewer for the comments. We have improved the background, significance of the study, discussion and conclusions as recommended.

Refer to Line: 31-33, 40, 46-51, 64-66, 68-72, 155-159, 182-189, 210-212.

Reviewer 2 Report

The study aims to provide an update on the prevalence of alpha-thalassemia in southeast Asia. The topic is commonly investigated, although not in the regions analyzed by the authors. The analysis was conducted properly and I believe that the manuscripts should be accepted by IJERPH however several major points must be addressed by the authors:

  • According to instruction for authors, the abstract should be structured but without headings. Please remove the headings
  • In materials and methods please provide the detailed PRISMA guidelines for conducting the analysis
  • Figure 1 has very poor quality, please provide a figure with higher quality
  • In supplement please provide all literature records analyzed in the study (242 papers), also from the initial search, not used in the final analysis
  • Please provide the detailed, step by step description of statistical analyses
  • Page 4, line 106, was should be were
  • Table 1 – what is events? What is total? Please elaborate
  • Figures 2, 3 and 4 – again low quality, please replace
  • In the table there are some lines without explanation, I understand that they are summarizing the results from each country, but please provide an explanation
  • The discussion is limited to several statements, please expand it

Reviewer 3 Report

Critique

This reviewer is a hematologist and geneticist and I cannot on comment on the appropriateness of using I2 index, Q-test, forest plot, Funnel plots and Egger’s tests and other population methods.

However, to define the overall prevalence of alpha thalassemia in different part of Southeast Asia is of importance and attempt to quantitative relative risk of lethal of hydrops fetalis and symptomatic hemoglobin H disease is a commendable effort. Unfortunately, as written it is not clear how the alpha thalassemia was determined at each analyzed paper and more importantly what alpha thalassemia types were encountered and their relative allelic frequencies.

The analysis then should be reevaluated based on the factors outlined in more details below:

The way Abstract is written starting with hydrops fetalis (homozygosity for a globin  --/-- genotype) then quoted prevalence of 55% in Vietnam would imply heterozygosity for --/aa which is unrealistic and would result in far greater number of hydrops fetalis than is observed. It is also of relevance to estimate relative risk for a compound alpha thalassemia heterozygosity of aa /-- and aa / a- genotypes resulting in the symptomatic hemoglobin H disease. Further, the genotypes of non-deletional alpha thalassemias such as hemoglobin Constant Spring and others further complicate the prediction of symptomatic and clinically relevant alpha thalassemias from far more common, clinically irrelevant, alpha thalassemia genotypes. It should be defined in Abstract and in more detail in Introduction that there are many alpha thalassemia phenotypes and genotypes in South East Asia and they highly vary in their frequency and genotypes, and also among different minorities in the same countries such as Vietnam. Thus, their analysis of each report should indicate which report did perform genotyping (a gold standard) and which was based on phenotyping by hemoglobin level, electrophoresis, and MCV which would miss large proportion of globin alpha genotype aa/a- (silent alpha thalassemia carrier). This is a condition which an alpha globin deletion is associated with single missing alpha globin gene, present in ~30% African Americans and its heterozygosity and homozygosity is of no clinical relevance. Analysis of reports from each individual country should also include what population was studied (an average citizen or a specific ethnic minority) and detailed prevalence of each genotype.

Round 2

Reviewer 2 Report

The paper have been improved and gained quality.

Author Response

Thank you for your comments.